# Online Rounding and Learning Augmented Algorithms for Facility Location

**Silvio Lattanzi**
Google Research
Barcelona, Spain
silviol@google.com

**Debmalya Panigrahi** *
Duke University
Durham, United States
debmalya@cs.duke.edu

**Ola Svensson** [†]
EPFL
Lausanne, Switzerland
ola.svensson@epfl.ch

## Abstract

Facility Location is a fundamental problem in clustering and unsupervised learning. Recently, significant attention has been given to studying this problem in the classical online setting enhanced with machine learning advice. While (almost) tight bounds exist for the fractional version of the problem, the integral version remains less understood, with only weaker results available. In this paper, we address this gap by presenting the first online rounding algorithms for the facility location problem, and by showing their applications to online facility location with machine learning advice. Beyond its implications for the learning augmented setting, our results also show that the hardness of the classic online facility location problem lies in computing a good fractional solution and not in rounding it.

## 1 Introduction

Clustering is a central problem in unsupervised learning. In recent years, to capture the evolving nature of real world data, there has been increased interest in clustering problems in the online setting, where the set of points that have to be clustered is not known in advance and is revealed to the algorithm over time Meyerson (2001); Alon et al. (2006); Fotakis (2011); Lattanzi and Vassilvitskii (2017); Almanza et al. (2021); Cohen-Addad et al. (2021); Fotakis et al. (2021b); Lattanzi et al. (2021); Anand et al. (2022); Cohen-Addad et al. (2022). This increased interest reflects the growing importance of performing learning tasks in uncertain and dynamically changing environments. To mitigate the negative impact of the uncertainty on algorithmic performance, a new paradigm using machine-learned predictions has rapidly gained traction in the last few years Purohit et al. (2018); Antoniadis et al. (2020); Bamas et al. (2020); Lattanzi et al. (2020); Wei and Zhang (2020); Im et al. (2021); Lykouris and Vassilvitskii (2021); Chen et al. (2022); Mitzenmacher and Vassilvitskii (2022); Bai and Coester (2023). The basic tenet of this framework is to simultaneously ensure that the algorithm is able to exploit *good* predictions about the future (called *consistency*) while being minimally affected by *bad* ones (called *robustness*), although it is unable to distinguish the good from the bad predictions presented to it. In this paper, we present new algorithms for the classical facility location problem in the online setting augmented with machine-learned predictions. These results are derived via novel online rounding algorithms of fractional solutions, which may be of independent interest.

In clustering problems, the goal is to partition (or cluster) a set of data points (called clients) into designated groups (or clusters) while optimizing a desired objective. One of the most popular is the $k$-median problem, which asks to select $k$ cluster centers such that the sum of distances of the clients from their nearest cluster centers is minimized. The condition that only $k$ centers can be chosen can be rather constraining, particularly in the online setting where it renders the problem uninteresting because the algorithm fails to remain competitive after it has opened all $k$ centers.

A natural and well-studied (Lagrangian) relaxation of $k$-median is the facility location problem, where an arbitrary number of centers (called facilities) can be opened but each open facility adds an

---

* Supported in part by NSF grants CCF-2329230 and CCF-1955703.

† Supported by the Swiss State Secretariat for Education, Research and Innovation (SERI) under contract number MB22.00054.

opening cost to the objective. The online facility location problem where clients arrive over time was introduced by Meyerson Meyerson (2001), and has since been studied extensively Anagnostopoulos et al. (2004); Fotakis (2007; 2008; 2011). The advantage of this setting is that it produces stable clusters that can be used in downstream tasks, notably as input to machine learning models, whereas changes to the clustering would incur substantial overhead. In recent years, interest has grown in obtaining learning-augmented algorithms for this problem, where machine-learned suggestions about the clustering solution are incorporated into the decision-making of the algorithm Almanza et al. (2021); Jiang et al. (2022); Fotakis et al. (2021a); Anand et al. (2022). This has led to almost tight bounds for *fractional* versions of this problem, while the integral version remains less understood, especially in the presence of multiple advice. In this paper, we address this gap by presenting the first online rounding algorithm for facility location, and show its applications to learning-augmented versions of the problem.

**Our Contributions.** We present two new online rounding algorithms for the facility location problem. Both algorithms take as input a fractional solution and produce an integral solution in the online setting. To the best of our knowledge, these are the first two online rounding algorithms for the metric facility location problem. This work introduces novel techniques that heavily exploit the problem's underlying metric structure, contributing to the growing literature on online rounding algorithms. We believe these techniques are also of interest for a broader range of clustering problems.

Our first algorithm is for the uniform version, where all facilities have the same opening cost. In this case, we give a deterministic online rounding algorithm that produces an integral solution whose cost is only a constant times that of the fractional solution input to it. Paired with prior results for the fractional problem, we obtain new algorithms for the *integral* facility location problem with machine learning advice that matches the almost tight results, which were previously only known for the fractional version.

Our second algorithm is for the non-uniform problem with arbitrary facility opening costs. For this more general problem, we give an online rounding algorithm that is randomized and loses a factor of $O(\log \log \Delta)$ in expectation compared to its fractional input, where $\Delta$ is the aspect ratio of the underlying metric space. We remark that, by standard techniques (see Appendix E), the upper bound $O(\log \log \Delta)$ may be replaced by $O(\log \log n)$, where $n$ is the number of clients in the instance. As in the uniform case, this algorithm can be paired with existing algorithms for the corresponding fractional problem and yields a nearly optimal competitive algorithm for the online problem.

As an application of our rounding algorithms, we obtain the *first* integral algorithm for online facility location in the multiple predictions setting Almanza et al. (2021); Anand et al. (2022). The consistency bounds that we obtain are tight up to lower-order terms, thereby bridging fractional and integral results for this problem. Simultaneously, we also obtain tight robustness bounds for the learning-augmented setting by using a combiner algorithm that obtains the better of the solutions between our algorithm and online facility location without predictions.

All our rounding algorithms run in polynomial time in the number of revealed clients; a detailed running time analysis is provided in Appendix F.

**Other Related Work.** The two most related works are Almanza et al. (2021); Anand et al. (2022). The first paper studied the learning-augmented facility location problem in the uniform case. Their integral algorithm has a cost overhead of $O(\log \log n)$ compared to the best fractional solution, and works in the simplified setting where all the predictions are presented before any client arrives. In comparison, we improve the cost overhead to $O(1)$ and no longer require the predictions to be given upfront. The second paper studied online covering problems with multiple machine learning advice. They provide several interesting results in this setting and in particular an online fractional algorithm to the learning-augmented facility location problem. Our work bridges this fractional result and the integral facility location problem. We also note that there is prior work on online facility location with a *single* machine learning advice Azar et al. (2022); Fotakis et al. (2025), or from a mechanism design perspective Balkanski et al. (2024). The results in these papers do not have any implication for the multiple predictions settings of the current paper, and are obtained using very different techniques.

More broadly, there has been significant recent interest in online rounding algorithms, especially for matching problems Buchbinder et al. (2023); Naor et al. (2025). In addition to its application to learning-augmented algorithms, our work adds to this portfolio of online rounding algorithms, specifically extending it beyond packing to covering problems. Our techniques differ from previous

work in that they use structural properties of the underlying metric space to define rounding solutions, both using deterministic and randomized tools. Indeed, this is in sharp contrast to prior work on online rounding for facility location in *non-metric* settings Alon et al. (2006); Bienkowski et al. (2020). In these papers, the rounding algorithms *must* incur a logarithmic loss (being basically identical to the set cover problem) while we incur sub-logarithmic loss by exploiting metric properties.

**Organization.** We formally define our problem and state our results in Section 2. In Section 3, we present the deterministic rounding algorithm for the uniform setting. In Section 4, we present the randomized rounding algorithm for the (general) non-uniform case. In Section 5, we give applications of these rounding algorithms to obtain new results for the learning-augmented facility location problem. In Appendix D, we present a lower bound showing that our analysis of the randomized rounding algorithm for the non-uniform setting is (asymptotically) tight. Finally, in Appendix F, we provide a running time analysis of our rounding methods.

## 2 PRELIMINARIES

**Online Facility Location.** The classic online facility location problem is defined on a metric space $(V, d)$, where $V$ is a set of vertices and $d$ is the distance function between vertex pairs satisfying the standard properties of a metric space: (i) (Non-negativity) $d(u, v) \geq 0$ for all $u, v \in V$ with $d(v, v) = 0$ for all $v \in V$, (ii) (Symmetry) $d(u, v) = d(v, u)$ for all $u, v \in V$, and (iii) (Triangle Inequality) $d(u, v) + d(v, w) \geq d(u, w)$ for all $u, v, w \in V$.

The metric space $(V, d)$ is revealed online to the algorithm: in each online step $t$, a new vertex $v_t \in V$ is revealed along with (i) its distance to all vertices revealed in previous steps, and (ii) its (non-negative) facility opening cost $c(v_t)$. If the opening cost is uniform across all vertices, then it is a *uniform* instance, otherwise it is a *non-uniform* instance. By scaling, all opening costs are 1 in the uniform case.

We denote the set of vertices revealed in the first $t$ steps $V_t$. In step $t$, the online algorithm's solution comprises a subset of vertices in $V_t$ where facilities are opened by the algorithm; we denote this set $F_t$. The sets $F_t$ must be monotone over time, i.e., an open facility cannot be closed: $F_1 \subseteq F_2 \subseteq \ldots \subseteq F_t$. The *connection cost* of a vertex $v \in V_t$ is its distance to the closest open facility. Collectively, the cost of the solution is the sum of the opening costs of the facilities and the connection costs of all vertices, i.e., $\mathsf{alg}_t = \sum_{v \in F_t} c(v) + \sum_{u \in V_t} \min_{v \in F_t} d(u, v)$.

Let $\mathsf{opt}_t$ denote the cost of an optimal solution on the instance $(V_t, d)$. Then, the competitive ratio of the online algorithm is defined as: $\max_t \mathsf{alg}_t / \mathsf{opt}_t$. Moreover, if the algorithm is randomized, the competitive ratio is $\max_t \mathbb{E}[\mathsf{alg}_t] / \mathsf{opt}_t$, where the expectation is over the random choices of the algorithm.

**Fractional solution and ML-advice.** To define a valid fractional solution, it would be convenient to first write a (standard) LP relaxation of the problem. The following is the LP at step $t$:

$$\text{minimize} \sum_{v \in V_t} c(v) \, y_v^t + \sum_{u \in V_t} \sum_{v \in V_t} d(u, v) \, x_{uv}^t \text{ such that}$$

$$\sum_{v \in V_t} x_{uv}^t = 1 \qquad \qquad \forall u \in V_t \qquad (1)$$

$$x_{uv}^t \leq y_v^t \qquad \qquad \forall v \in V_t \qquad (2)$$

$$x_{uv}^t, y_v^t \geq 0 \qquad \qquad \forall u, v \in V_t \qquad (3)$$

The online fractional solution in step $t$ is a feasible solution to this LP. Moreover, the variables $y_v^t$ that represent the fraction of the facility at vertex $v$ that is open at time $t$ are non-decreasing over time: $y_v^1 \leq y_v^2 \leq \ldots$ for all $v$. We will refer to the value of $y_v^t$ as the *fractional mass* at vertex $v$ at time $t$.

Note that the values of $y_v^t$ completely specify the fractional solution even without explicitly defining $x_{uv}^t$. This is because the optimal values of the $x_{uv}^t$ variables is given as follows: for each client $u \in V_t$, order the vertices in $V_t$ in non-decreasing distance from $u$ (breaking ties arbitrarily), and select the minimal prefix of this order such that the total fractional mass in this prefix is at least 1. Now, assign $x_{uv}^t = y_v^t$ for all facilities $v$ in this prefix, except possibly for the last one. For this last

vertex, the value of $x_{uv}^t$ is such that sum of $x_{uv}^t$ over all the facilities in the prefix is 1. The value of $x_{uv}^t$ for all vertices outside this prefix is 0.

In the learning-augmented facility location problem, whenever a new vertex $v$ arrives, one also receives $k$ feasible suggestions (predictions) $y_v(1), \ldots, y_v(k)$ for the value of $y_v$. We consider the $k$ suggestions at each step as a collection (or bag) of predictions, disregarding any association with specific predictors. Our objective is to achieve performance comparable to the best suggestion in each step, i.e., we seek the minimum-cost solution that is consistent with at least one suggestion at every stage. Formally,

$$\mathsf{dynamic}_t = \min_{\hat{y} \in \hat{Y}} \sum_{v \in V_t} c(v)\, \hat{y}_v + \sum_{u \in V_t} \sum_{v \in V_t} d(u,v)\, x_{uv}^t,$$

where $\hat{Y} = \{\hat{y} : \forall v \in V_t, \exists i \in [k], \hat{y}_v = y_v(i)\}$ and $x_{uv}^t$ are defined using the values of $\hat{y}$ as described above.

It is interesting to note that this model of predictions generalizes other natural types of predictions. E.g., if the predictions specify opening a facility at a vertex when it arrives or even at the very outset Almanza et al. (2021), it can be simulated by setting $y_v = 1$ in our prediction model. In fact, our prediction model captures the very natural setting where the variable $y_v$ represents the probability that a facility is opened at vertex $v$.

We also note that our algorithm consider the possibility to open previously specified facilities as long as the assignment is online. This is standard in the literature. In fact, in the traditional online model for the facility location problem, introduced by Meyerson (2001), for non-uniform facility location(this is specified in the first paragraph of Section 3 in Meyerson (2001)) the facility locations are known before the clients arrive online, the demands are specified online and the assignments and the opening are decided online (in particular any facility location can be opened at any time and this is necessary in this setting). Furthermore, in Almanza et al. (2021), in the non-uniform case, the set of suggested facilities is specified in advance and facilities can be opened in these specified locations.

In the learning-augmented facility location problem, we aim to return an integral solution (i.e. a solution where all the $y$ variables have integral value) such that its cost is bounded by $\min\{\alpha\, \mathsf{dynamic}_t, \beta\, \mathsf{opt}_t\}$ for $\alpha$ and $\beta$ as small as possible. From lower-bounds in previous works Almanza et al. (2021); Anand et al. (2022), we know that $\alpha \geq \frac{\log k}{\log \log k}$ and that $\beta \geq \frac{\log t}{\log \log t}$ Fotakis (2008).

With the above notation, we can state the implication of our rounding algorithms more formally for the learning-augmented problems. First, we study the uniform setting where all the facilities have the same opening cost. In this setting, our rounding algorithm implies a deterministic algorithm that obtains a $O(\min\{\log(k+1)\, \mathsf{dynamic}_t, \frac{\log t}{\log \log t}\, \mathsf{opt}_t\})$-approximation, improving previous work in the area. Second, we use our non-uniform facility rounding algorithm to obtain the *first* learning-augmented algorithm for the non-uniform facility location problem. Our algorithm returns a $O(\log \log \Delta \cdot \min\{\log(k+1)\, \mathsf{dynamic}_t, \frac{\log t}{\log \log t}\, \mathsf{opt}_t\})$-approximation(recall that, by standard techniques (see Appendix E), the upper bound $O(\log \log \Delta)$ may be replaced by $O(\log \log n)$). Both results start from the fractional solution built for the learning-augmented facility location problem designed in previous work Anand et al. (2022) and use our online rounding algorithms to obtain the final integral results.

**Additional notation.** Consider the metric space $(V_t, d)$ at time $t$. The ball centered at some vertex $v \in V_t$ with radius $R \geq 0$ in this metric space is denoted $B^t(v, R)$: $B^t(v, R) = \{u \in V_t : d(u,v) \leq R\}$. For any set of vertices $S \subseteq V_t$, the total fractional mass on the vertices of $S$ at time $t$ is denoted $y^t(S)$: $y^t(S) = \sum_{v \in S} y_v^t$. Clearly, $y^t(S)$ is also non-decreasing over time. In particular, we will often consider the total fractional mass at time $t$ in a ball $B = B^t(v, r)$; this is denoted $y^t(B)$.

**$\gamma$-Consistent Rounding.** Consider a fractional solution $\mathbf{y}^t = (y_v^t, v \in V_t)$ (recall from above that $\mathbf{y}^t$ is sufficient to define a solution). Now, suppose we round this fractional solution to produce an integer solution $F_t \subseteq V_t$. We say that $F_t$ is $\gamma$-consistent with $\mathbf{y}^t$ if the following property holds:

($\gamma$-Consistency.) For any ball $B = B^t(v, R)$ centered at a vertex $v \in V_t$ and with radius $R$, if $y^t(B) \geq {}^1\!/{}_2$, then there is an open facility in $F_t$ that is within distance $\gamma R$ of $v$.

We show that the $\gamma$-consistency property implies that the connection costs of the fractional and integral solutions are related.

**Lemma 1.** *If an integral solution $F_t$ is $\gamma$-consistent with a fractional solution $\mathbf{y}^t$, then the total connection cost of all the clients in $V_t$ in the integer solution is at most $2\gamma$ times that in the fractional solution. In notation,*

$$\sum_{u \in V_t} \min_{v \in F_t} d(u, v) \leq 2\gamma \cdot \sum_{u \in V_t} \sum_{v \in V_t} d(u, v) \, x_{uv}^t,$$

*for any $\mathbf{x}^t$ such that $(\mathbf{y}^t, \mathbf{x}^t)$ is feasible for the LP given above.*

*Proof.* Consider a client at vertex $u$ that arrives at time $t$. Suppose its fractional connection cost is $\beta$, i.e., $\sum_{v \in V} d(u, v) x_{uv}^t = \beta$ where $\sum_{v \in V} x_{uv}^t = 1$ and $x_{uv}^t \leq y_v^t$ for all $v \in V_t$. Then, $\sum_{v \in B(u, 2\beta)} y_v^t \geq \sum_{v \in B(u, 2\beta)} x_{uv}^t \geq 1/2$. Note that $\gamma$-consistency ensures that there is at least one facility at distance $2\gamma\beta$ from $u$ after the processing at time $t$. It follows that the connection cost of client $u$ in the integer solution is at most $2\gamma\beta$. $\qquad\square$

Our goal is to obtain $\gamma$-competitive solutions so that the total connection cost can be bounded immediately by Lemma 1. So, our analysis will comprise two parts: first, obtain an explicit bound on the facility opening cost $\sum_{v \in F_t} c(v)$ of the integral solution against the opening cost of the fractional solution, $\sum_{v \in V_t} c(v) \, y_v$; and second, establish $\gamma$-competitiveness of the integral solution with respect to the fractional solution for a suitable value of $\gamma$.

## 3  DETERMINISTIC ROUNDING ALGORITHM FOR UNIFORM OPENING COSTS

In this section we present our deterministic online rounding algorithm for the uniform facility case. This algorithm combined with the fractional algorithm in Anand et al. (2022) will imply our result for learning-augmented facility location as we will show in Section 5.

We start by stating the main result of this section.

**Theorem 2.** *There exists an online algorithm that rounds a fractional solution of uniform facility location online and the cost of the returned integral solution is $O(\alpha)$, where $\alpha$ denotes the cost of the fractional solution.*

**The Algorithm.** The main idea behind the algorithm is to guarantee that at any point in time the algorithm is 4-consistent. In order to do so if at time $t$ a ball $B^t(v, R)$ has total fractional mass $y^t(B) \geq 1/2$ and has the closest open facility at a distance $\geq 4R$ we open a facility at $v$. While this guarantees 4-consistency, it can lead to excessive facility openings. To mitigate this, additional facilities are opened. Roughly, if there is a ball $B^t(u, r)$ with total fractional mass in the ball $y^t(B) \geq 1/4$ that intersects $B^t(v, R)$ and such that the closest open facility to $u$ is at a distance $\geq 3r$ we open also a facility at $u$. In addition, we do this an additional time by considering balls $B^t(w, \rho)$ with total fractional mass in the ball $y^t(B) \geq 1/8$, intersecting with $B^t(u, r)$ and with closest open facility to $w$ is at a distance $> 2\rho$ and by opening a facility in such $w$. Interestingly, by opening these additional facilities we can show that the number of open facility is bounded and by combining this with the fact that the algorithm is 4-consistent we obtain the theorem.

Roughly speaking the algorithm uses a three levels covering scheme. Level-1 (Condition A balls): The algorithm identifies a large primary region, $B(v, R)$, containing significant fractional mass ($\geq 1/2$) that lacks nearby facilities (distance $> 4R$). It commits to opening a facility here. Level-2 (Condition B balls): To ensure dense, off-center subsets of this demand are adequately served, the algorithm searches for smaller overlapping balls, $B(u, r)$, with a mass of $\geq 1/4$ that are also poorly served (distance $> 3r$). Level-3 (Condition C balls): Finally, it recursively checks for even smaller, isolated pockets of mass ($\geq 1/8$) overlapping the Level-2 ball. Crucially, to prevent over-opening, these Level-3 balls are proven to be mutually disjoint, meaning a Level-2 ball will spawn at most 3 Level-3 facilities.

To define the algorithm, we start to formalize the three conditions discussed above:

- A ball $B = B^t(v, R)$ is said to satisfy **Condition A** at time $t$ if the following holds:
  - the total fractional mass in the ball $y^t(B) \geq 1/2$
  - the closest open facility to $v$ is at a distance $> 4R$.

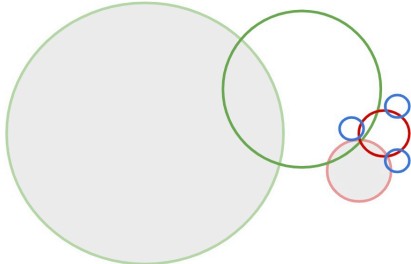

Figure 1: Green balls are at level-1, red balls are at level-2, blue balls are at level-3. Level-3 balls associated with the same level-2 balls are disjoint and are at most 3. Level-2 and level-1 balls can overlap but any new ball is associated with some new additional fractional mass. This allows us to bound the number of level-2 and level-1 balls.

- A ball $B = B^t(u, r)$ is said to satisfy **Condition B** at time $t$ if the following holds:
  - the total fractional mass in the ball $y^t(B) \geq 1/4$
  - the closest open facility to $u$ is at a distance $> 3r$.

- A ball $B = B^t(w, \rho)$ is said to satisfy **Condition C** at time $t$ if the following holds:
  - the total fractional mass in the ball $y^t(B) \geq 1/8$
  - the closest open facility to $w$ is at a distance $> 2\rho$.

We use the above conditions to define the online algorithm. In the description below, we define the algorithmic steps at some time $t$. For the purpose of analysis, it would be convenient to think of time as continuous, although the actual implementation of the algorithm can be easily discretized by skipping over times where the online algorithm does not make any updates.

When a client arrives, the algorithm first updates the fractional solution, then rounds it based on Algorithm 1, and finally connects the client to the closest open facility.

---

**Algorithm 1:** The Deterministic Online Rounding Algorithm for Uniform Facility Location

---

**while** $\exists$ *a ball* $B(v, R)$ *satisfying condition A* **do**

    Let $B(v, R)$ be a ball of minimum radius satisfying the condition A.

    **while** $\exists$ *a ball* $B(u, r)$ *for some* $r \leq R/3$ *satisfying condition B s.t.* $B(u, r) \cap B(v, R) \neq \emptyset$ **do**

        Let $B(u, r)$ be a ball of minimum radius satisfying the condition B and intersecting $B(v, R)$;

        **while** $\exists$ *a ball* $B(w, \rho)$ *for some* $\rho \leq r/2$ *satisfying condition C s.t.* $B(w, \rho) \cap B(u, r) \neq \emptyset$ **do**

            Let $B(w, \rho)$ be a ball of minimum radius satisfying the condition C and intersecting $B(u, r)$;

            Open a facility at $w$;

        Open a facility at $u$;

    Open a facility at $v$;

---

**Analysis.** First, we bound the facility cost, namely we show that the number of facilities opened by the algorithm is at most a constant times the final value of the fractional solution $\sum_v y_v$.

We call the ball selected in the outer loop level-1 ball and the respective facility level-1 facility, the one selected in the second loop level-2 ball and level-2 facility and the one selected in the innermost loop level-3. In Figure 1, we show graphically the interaction between ball at different levels. Note that a ball at level-1 only open a single in the outer loop, a ball at level-2 open a single facility at level-2 but possibly multiple facilities at level-3. So in our proof we first bound the number of facilities opened in level-3. Then we show how to bound the number of level-1 and level-2 facilities. To do this, we first need to establish disjointness of the level-3 balls associated with a level-2 ball.

**Lemma 3.** *The level-3 balls associated with a level-2 ball $B(u, r)$ are mutually disjoint.*

*Proof.* Suppose not; let $B(w, \rho)$ and $B(w', \rho')$ overlap. Without loss of generality, suppose $\rho \leq \rho'$. Then, the algorithm processes $B(w, \rho)$ before $B(w', \rho')$. At the end of the processing for $B(w, \rho)$, the algorithm opens a level-3 facility at $w$. Since $B(w, \rho)$ and $B(w', \rho')$ overlap, we have $d(w, w') \leq \rho + \rho'$. Therefore, there is an open facility at distance $\leq \rho + \rho'$ from $w'$. Yet, $B(w', \rho')$ satisfies condition C; this implies $\rho + \rho' > 2\rho'$, i.e., $\rho > \rho'$. This is a contradiction. $\qquad\square$

**Lemma 4.** *The total number of level-3 facilities opened by a (level-2) ball $B(u, r)$ is at most 3.*

*Proof.* Suppose $B(u, r)$ opens $\geq 4$ level-3 facilities. Since the level-3 balls corresponding to these facilities are mutually disjoint (by Lemma 3), it follows that the total fractional mass in these level-3 balls is at least $4 \cdot 1/8 = 1/2$. Now, note that the ball $B(u, 2r)$ contains all these level-3 balls; hence, this ball also has fractional mass at least $1/2$. The radius of this ball is $2r \leq 2R/3 < R$. Since this ball has a smaller radius than $B(v, R)$ but $B(v, R)$ was chosen as a minimum radius ball satisfying condition A, it must be the case that $B(u, 2r)$ does not satisfy condition A. Since $y^t(B(u, 2r)) \geq 1/2$, the only way it can fail to satisfy condition A is if there is an open facility at distance $\leq 8r$ from $u$. But, since the balls $B(v, R)$ and $B(u, r)$ overlap, $d(v, u) \leq R + r$. Thus, $v$ has an open facility at distance at most $8r + (R + r) \leq 8R/3 + (R + R/3) \leq 4R$. This contradicts the fact that the ball $B(v, R)$ satisfies condition A. $\qquad\square$

Next, we bound the number of level-1 and level-2 facilities. The arguments for these two cases are quite similar. We start by defining predecessors for level-1 and level-2 facilities. Consider a level-1 facility at vertex $v$ opened by a level-1 ball $B(v, R)$ at time $t$. We call a level-1 facility at some vertex $v'$ the *predecessor* of the level-1 facility at $v$ if the following holds:
– The level-1 facility at $v'$ was opened before the level-1 facility at $v$.
– The level-1 ball $B(v', R')$ that opened the level-1 facility at $v'$ overlaps with the level-1 ball $B(v, R)$.
– The level-1 balls corresponding to the level-1 facilities opened after $v'$ and before $v$ do not overlap with the level-1 ball $B(v, R)$.

**Lemma 5.** *Consider a level-1 facility opened by a ball $B(v, R)$ at time $t$. Let the predecessor of this level-1 facility at $v$ be a level-1 facility at $v'$ that was opened by the level-1 ball $B(v', R')$ at time $t' \leq t$. Then,*

$$y^t(B(v, R)) - y^{t'}(B(v, R)) \geq 1/4.$$

*Proof.* Since the balls $B(v, R)$ and $B(v', R')$ overlap, we have $d(v, v') \leq R + R'$. But, note that the ball $B(v, R)$ satisfied condition A at time $t$, i.e., there was no open facility within a distance of $4R$ of $v$ at time $t$. In particular, this implies that the facility at $v'$ is also at a distance $> 4R$ from $v$, namely $d(v, v') > 4R$. It follows that $R + R' > 4R$, i.e., $R' > 3R$.

Suppose the lemma is false, i.e., $y^t(B(v, R)) - y^{t'}(B(v, R)) < 1/4$. Since $y^t(B(v, R)) \geq 1/2$, it follows that $y^{t'}(B(v, R)) > 1/4$. Since the processing of $B(v', R')$ did not open a facility at $v$, it must be that $B(v, R)$ failed condition B at the end of this processing. Since $y^{t'}(B(v, R)) > 1/4$, the only way this can happen is if there was a facility open at distance $\leq 3R$ from $v$ at the end of the processing for $B(v', R')$. But, this contradicts the fact that $B(v, R)$ satisfied condition A at time $t$. $\qquad\square$

The definition of the predecessor of a level-2 facility is similar to that for level-1 facilities. Consider a level-2 facility at vertex $u$ opened by a level-2 ball $B(u, r)$ at time $t$. We call a level-2 facility at some vertex $u'$ the *predecessor* of the level-2 facility at $u$ if the following holds:
– The level-2 facility at $u'$ was opened before the level-2 facility at $u$.
– The level-2 ball $B(u', r')$ that opened the level-2 facility at $u'$ overlaps with the level-2 ball $B(u, r)$.
– The level-2 balls corresponding to the level-2 facilities opened after $u'$ and before $u$ do not overlap with the level-2 ball $B(u, r)$.

Now, using the same proof strategy of Lemma 5, one can show that the overlap between level-2 ball is bounded. Here we present the statement of the Lemma and defer the proof to Appendix A

**Lemma 6.** *Consider a level-2 facility opened by a ball $B(u, r)$ at time $t$. Let the predecessor of this level-2 facility at $u$ be a level-2 facility at $u'$ that was opened by the level-2 ball $B(u', r')$ at time $t' \leq t$. Then,*

$$y^t(B(u, r)) - y^{t'}(B(u, r)) \geq 1/8.$$

We are now ready to bound the facility cost:

**Theorem 7.** *The number of facilities opened by the rounding algorithm is at most* $36 \cdot \sum_v y_v$, *where* $y_v$ *is the final value of the fractional solution for vertex* $v$.

*Proof.* We charge a level-1 facility opened by a level-1 ball $B(v, R)$ to the gain in fractional mass $y(B(v, R))$ after time $t'$ until time $t$, where $t'$ denotes the time when the predecessor of the facility at $v$ was opened. This charging only loses a factor of $4$ by Lemma 5.

We charge the level-2 facility and the level-3 facilities opened by a level-2 ball $B(u, r)$ to the gain in fractional mass $y(B(u, r))$ after time $t'$ until time $t$, where $t'$ denotes the time when the predecessor of the facility at $u$ was opened. The charging of the level-2 facility only loses a factor of $8$ by Lemma 6 and that of the level-3 facilities loses a factor of $24$ by Lemma 4.

The lemma follows by adding up the multiplicative factors that we lose for level-1, level-2, and level-3 facilities. $\qquad\square$

Finally, we bound the connection cost.

**Theorem 8.** *Let $U$ be the set of clients and $x_{uv}$ be the fractional solution for client $u \in U$. Then, the connection cost of integer solution for client $u$ is at most* $8 \sum_{v \in V} d(u, v) x_{uv}$.

*Proof.* This follows directly from the fact that the algorithm is $4$-consistent, via Lemma 1. $\qquad\square$

Now we are ready to prove Theorem 2.

*Proof of Theorem 2.* The proof follow by combining Theorem 7 and Theorem 8. $\qquad\square$

## 4 RANDOMIZED ROUNDING ALGORITHM FOR GENERAL OPENING COSTS

In this section, we describe a randomized rounding algorithm for the weighted online facility location problem with the following properties: (a) the rounding algorithm is $\gamma$-consistent for some constant $\gamma$ (this property holds deterministically), and (b) at any time $t$, the *expected* facility opening cost of the integral solution is at most $O(\log \log \Delta_t)$ times that of the fractional solution, where $\Delta_t$ denotes the aspect ratio of the metric space at time $t$, i.e., the ratio of the maximum to the minimum (non-zero) distance between any pair of vertices. Formally,

**Theorem 9.** *There exists a randomized online algorithm that rounds a fractional solution for facility location online and the expected cost of the rounded integral solution is $O((\log \log \Delta) \cdot \alpha)$, where $\alpha$ denotes the cost of the fractional solution.*

In defining the rounding algorithm, it will be convenient to assume that the fractional mass $y_v$ at any vertex $v$ does not change over time. This is without loss of generality by the following standard technique: whenever the fractional algorithm increases the fractional mass at a vertex $v$ by some quantity $\delta$, we create a second copy of vertex $v$, namely a new vertex that is at distance 0 from $v$ and at the same distance as $v$ from all other vertices, and set the fractional mass on the new vertex to $\delta$. The facility opening cost of the new vertex is the same as that of the original vertex. Clearly, the opening and connection costs of the new fractional solution are identical to the original solution, but this new solution has the advantage that the fractional mass on a vertex is set in only one step. In the rest of this paper, we will assume that the fractional solution satisfies this property. We emphasize that this is not an additional assumption on the input; rather, it is a purely notational transformation that preserves all opening and connection costs and works for any metric space. Moreover, this transformation does not affect the running time of the algorithm beyond constant factors.

We describe the rounding algorithm next, and then give its analysis in Appendix B to establish properties (a) and (b) above.

## 4.1 Randomized Rounding Algorithm

If any vertex $v$ has $y_v^t \geq 1/2$, we immediately open a facility at that vertex. We call this a *deterministic* rounding step, and the corresponding facilities are called deterministic facilities. Clearly, the total opening cost of deterministic facilities is at most twice their fractional opening cost. In the rest of the description, we focus on how the algorithm opens the rest of the facilities using a randomized algorithm.

The algorithm uses a counter for every vertex $v$ that we call its *level* and denote $\ell(v)$. In the following, we will say that a vertex has been rounded by the randomized algorithm if it has been involved in a randomized step of the algorithm. Initially, $\ell(v) = 0$. Over time, $\ell(v)$ tracks the number of times vertex $v$ has been rounded by the algorithm. Since the algorithm rounds vertices randomly, the level counters are random variables. Eventually, we will show that for every vertex $v$, the expected value of $\ell(v)$ at time $t$ is $O(\log \log \Delta_t)$. For notational convenience, we also maintain a level counter of value 1 at each deterministic facility. These counters do not change over time.

It will be convenient to maintain an order $\prec$ on all the vertices in the metric space. The precise order is not important, but for consistency, $v \prec v'$ if $v$ appears earlier than $v'$ in the online problem (ties are broken arbitrarily for vertices that appear in the same time-step). We also set up a lexicographic order on all balls $B^t(v, R)$ in the metric space, which we also denote $\prec$, using the following rules: (a) if $t < t'$, then $B^t(v, R) \prec B^{t'}(v', R')$ for any vertices $v, v'$ and radii $R, R'$, (b) for any time $t$, if $R < R'$, then $B^t(v, R) \prec B^t(v', R')$ for any vertices $v, v'$, and (c) for any time $t$ and radius $R$, if $v \prec v'$, then $B^t(v, R) \prec B^t(v', R)$.

At any time $t$, we call a ball $B^t(v, R)$ *critical* if it satisfies the following properties: (a) the total fractional mass in $B^t(v, R)$ is at least $1/2$, and (b) there is no overlapping critical ball with radius at most $2R$ that appears earlier in the lexicographic order, i.e., there is no $B^{t'}(v', R') \prec B^t(v, R)$ with radius $R' \leq 2R$ such that $B^{t'}(v', R') \cap B^t(v, R) \neq \emptyset$. Observe that critical balls are defined deterministically since this definition only depends on the fractional solution. We also define the level of a critical ball $B := B^t(v, R)$ as $\ell(B) := \min\{\ell : \sum_{u \in B : \ell(u) \leq \ell} y_u^t \geq 1/4\}$. Note that although the fact that a ball is critical is deterministic, its level is random since it depends on the values of the level counters which themselves are random variables.

Finally, we define the randomized rounding step. At time $t$, we consider the critical balls in lexicographic order $\prec$. For a critical ball $B := B^t(v, R)$, if there is an open facility in $B$ already, then we do nothing. Otherwise, we open a facility at a location in $B_\ell := \{u \in B : \ell(u) \leq \ell(B)\}$ with probability proportional to $y_u^t$. (In other words, the probability of opening a facility at $u \in B_\ell$ is $y_u^t / \sum_{u \in B_\ell} y_u^t$.) Correspondingly, we increase by one the level counters $\ell(u)$ of all locations $u \in B_\ell$. This algorithm is already sufficient, but for the sake of simpler analysis, we add one more step. For each vertex $u \in B_\ell$, we also open a facility at $u$ independently with probability $y_u^t$. We call the two randomized rounding steps respectively the *dependent* and *independent* rounding steps.

**Why a constant factor remains open in the non-uniform case.** We show in Appendix D that the analysis of our randomized algorithm is asymptotically tight. In particular, any improvement over the $O(\log \log \Delta)$ bound would require fundamentally new ideas. At a high level, the key structural difference between the uniform and non-uniform settings lies in how facility opening costs can be charged to fractional mass. In the uniform case, the cost of opening a facility at a vertex can be charged to fractional mass in nearby regions of the metric space. This charging argument crucially exploits the uniformity of opening costs. In contrast, in the non-uniform case, the opening cost of a facility at a vertex cannot, in general, be charged to fractional mass located elsewhere in the metric space.

## 5 Application to Learned-Augmented Facility Location

In this section we show how to leverage the presented online rounding algorithms to obtain algorithms for the learning-augmented facility location.

Our starting point is the online fractional algorithm for learning-augmented facility location presented in Anand et al. (2022) that gives the following theorem:

**Theorem 10** (Theorem 7.1 from Anand et al. (2022) restated)**.** *There is an algorithm for the fractional online facility location problem that at time $t$ produces an online solution with cost $O(\min\{\log(k+1)$ dynamic$_t, \frac{\log t}{\log \log t}$ opt$_t\})$ in the multiple predictions setting with $k$ predictions.*

We apply our rounding algorithms presented in Section 3 and Section 4 on the fractional solution constructed in Anand et al. (2022) to obtain integral algorithms for the learning-augmented facility location. In particular, we obtain the following theorems for the uniform (Theorem 11) and non-uniform (Theorem 12) setting. Note that the two theorems obtain consistency bounds of $O(\log(k+1))$ and $O(\log \log \Delta \cdot \log(k + 1))$ respectively, while ensuring a robustness bound of $O(\frac{\log t}{\log \log t})$. (The proofs are deferred to Appendix C. Also recall that, by standard techniques (see Appendix E), the upper bound $O(\log \log \Delta)$ may be replaced by $O(\log \log n)$.)

**Theorem 11.** *There is an algorithm for the uniform learning-augmented facility location problem that at time $t$ produces an online solution with cost $O(\min\{\log(k+1)$ dynamic$_t, \frac{\log t}{\log \log t}$ opt$_t\})$.*

**Theorem 12.** *There is an algorithm for the non-uniform learning-augmented facility location problem that at time $t$ produces an online solution with cost $O(\min\{\log \log \Delta \cdot \log(k + 1)$ dynamic$_t, \frac{\log t}{\log \log t}$ opt$_t\})$.*

## CONCLUSIONS AND FUTURE WORK

We present two new algorithms to round online a fractional facility location solution and we show how to use them to obtain learning-augmented facility location algorithms . The algorithms obtains almost tight guarantees for the learning-augmented problem and are simple and natural. As follow-up questions, it would be interesting to find a rounding algorithm for the non-uniform settings that lose only a constant factor in the approximation. It would also be very nice to modify the fractional algorithm in Anand et al. (2022) to obtain a $O(\min\{\frac{\log k}{\log \log k}$ dynamic$_t, \frac{\log t}{\log \log t}$ opt$_t\})$-approximation or show that it is impossible.

## IMPACT STATEMENT

This paper presents work whose goal is to advance the field of Machine Learning. The main contribution of the paper is theoretical and is of interest to the domain of designing robust algorithms leveraging machine learning advice.

## ACKNOWLEDGMENT

This research project was initiated at Dagstuhl Seminar 23061 "Scheduling." We thank the organizers and Dagstuhl for their support.

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
