*Proof.* Since the balls $B(u, r)$ and $B(u', r')$ overlap, we have $d(u, u') \leq r + r'$. But, note that the ball $B(u, r)$ satisfied condition B at time $t$, i.e., there was no open facility within a distance of $3r$ of $u$ at time $t$. In particular, this implies that the facility at $u'$ is also at a distance $> 3r$ from $u$, namely $d(u, u') > 3r$. It follows that $r + r' > 3r$, i.e., $r' > 2r$.

Suppose the lemma is false, i.e., $y^t(B(u, r)) - y^{t'}(B(u, r)) < 1/8$. Since $y^t(B(u, r)) \geq 1/4$, it follows that $y^{t'}(B(u, r)) > 1/8$. Since the processing of $B(u', r')$ did not open a facility at $u$, it must be that $B(u, r)$ failed condition C at the end of this processing. Since $y^{t'}(B(u, r)) > 1/8$, the only way this can happen is if there was a facility open at distance $\leq 2r$ from $u$ at the end of the processing for $B(u', r')$. But, this contradicts the fact that $B(u, r)$ satisfied condition B at time $t$. $\qquad \square$

## B  Analysis of the Randomized Rounding Algorithm

The $\gamma$-consistency property follows immediately from the definition of the algorithm.

**Lemma 14.** *The randomized rounding algorithm given above is 5-consistent.*

*Proof.* Let $B := B^t(v, R)$ be such that $y^t(B) \geq 1/2$. If $B$ is critical, then it contains an open facility after its rounding step. So, suppose $B$ is not critical. Then, there must be some critical ball $B' \prec B$ of radius at most $2R$ overlapping with $B$; call this ball $B' := B(v', R')$. Since the balls overlap and $R' \leq 2R$, we have $B' \subseteq B(v, 5R)$ by triangle inequality. After the rounding step for $B'$, there must be an open facility in $B'$, which is at a distance of at most $5R$ from $v$. This establishes the lemma. $\qquad \square$

We now bound the expected facility opening cost incurred by the rounding algorithm. Note that the level of a vertex $\ell(v)$ denotes the number of times it has been rounded.

**Lemma 15.** *Fix any vertex $v$ and let $y_v$ be the value of $y_v^t$ at the end of the algorithm. If $\ell(v)$ denotes the number of times that vertex $v$ is rounded in the entire algorithm, then the expected facility cost at $v$ is at most $5c(v)y_v \cdot \mathbb{E}[\ell(v)]$.*

*Proof.* First, note that if there is a deterministic facility at $v$, then the lemma holds for $v$ since the cost of the facility is $c(v)$, the level of the vertex $\ell(v) = 1$, and the fractional value $y_v \geq 1/2$. In the rest of the proof, we assume that $y_v < 1/2$, i.e., there is no deterministic facility at $v$. Next, note that $y_v^t = y_v$ at all times $t$ when $v$ is rounded, since $y_v$ is set in only one step of the fractional algorithm and $v$ is rounded only after $y_v^t > 0$. Moreover, the cumulative fractional mass of all vertices in $B_\ell$ for randomized rounding of a ball $B$ is at least $1/4$. I.e., the probability that a facility is opened at $v$ in a single dependent rounding step is at most $4y_v$. Including the independent rounding step, the total probability of opening a facility at $v$ is at most $5y_v$.

Consider the random variables $Y_i(v)$ with value $c(v)$ if a facility is opened at vertex $v$ when $\ell(v) = i$, and $0$ otherwise. These random variables are independent across different values of $i$, and their expected value is bounded by $\mathbb{E}[Y_i(v)] \leq 5c(v)y_v$ for every $i$. The number of such random variables is given by the final value of $\ell(v)$, which is determined by a stopping rule. Using Wald's identity, we then have that the expected facility cost at $v$ is at most $5c(v)y_v \cdot \mathbb{E}[\ell(v)]$. $\qquad \square$

By the above lemma, it suffices to bound the value $\mathbb{E}[\ell(v)]$ for any vertex $v$. Indeed, we will show that $\mathbb{E}[\ell(v)] \leq O(\log \log \Delta)$. But first, we establish a simpler deterministic bound on $\ell(v)$.

**Lemma 16.** *Any vertex $v$ belongs to at most $1 + \lg \Delta$ critical balls. Therefore, $\ell(v) \leq 1 + \lg \Delta$.*

*Proof.* Clearly, the lemma holds if there is a deterministic facility at $v$, since $\ell(v) = 1$. Hence, we assume $y_v < 1/2$ in the rest of the proof. Suppose $v$ is in two critical balls $B := B^t(u, R)$ and $B' := B^{t'}(u', R')$, where wlog, $t \leq t'$. Since these balls overlap, it must be that $R > 2R'$ by the definition of critical balls. Now, since $y_v < 1/2$, the minimum radius of any critical ball containing $v$ must be 1. The maximum radius of any ball in the metric space is $\Delta$. The lemma follows. $\square$

Next, we bound the expected level of a vertex. We do this in two steps. In the first step, we give a tail bound on the level of a critical ball that the algorithm performs randomized rounding on.

**Lemma 17.** *For any critical ball $B$, the probability that the algorithm performs the randomized rounding step for $B$ at level $\ell(B) > \ell$ for any positive integer $\ell$ is at most $e^{-\ell/4}$.*

*Proof.* If the algorithm performs randomized rounding for $B$, then it must be that all previous rounding attempts for vertices in $B$ did not open any facility. In particular, this is true for the independent rounding attempts on these vertices. Each independent rounding attempt for a vertex $v$ with fractional value $y_v$ fails with probability $1 - y_v \leq e^{-y_v}$. Since these rounding attempts are independent, the probability that all these attempts fail to open any facility is at most $e^{-\sum_{v \in B} y_v \cdot \ell(v)}$. Since $\ell(B) \geq \ell$ and $\sum_{v \in B} y_v \geq 1/2$ (the latter because $B$ is critical), we get $e^{-\sum_{v \in B} y_v \cdot \ell(v)} \leq e^{-\ell/4}$. $\square$

We now use this lemma to bound the expected level of any vertex.

**Lemma 18.** *The expected level of any vertex at the end of the algorithm is at most $1 + 8\ln(1 + \lg \Delta) = O(\log \log \Delta)$.*

*Proof.* Fix any vertex $v$. By Lemma 16, it belongs to at most $1 + \lg \Delta$ critical balls; call them $B_1, B_2, \ldots, B_k$ where $k \leq 1 + \lg \Delta$. For each such ball $B_i$, by Lemma 17, randomized rounding is performed for ball $B_i$ at a level $> 8\ln(1 + \lg \Delta)$ with probability at most $e^{-2\ln(1+\lg \Delta)} = 1/(1+\lg \Delta)^2$. Using the union bound over the $k \leq 1 + \lg \Delta$ balls, the probability that randomized rounding is performed for *any* ball containing $v$ at a level $> 8\ln(1 + \lg \Delta)$ is at most $1/1 + \lg \Delta$. Since the maximum value of $\ell(v)$ is (deterministically) $1 + \lg \Delta$ by Lemma 16, it follows that the expected value of $\ell(v)$ is at most $1 + 8\ln(1 + \lg \Delta) = O(\log \log \Delta)$. $\square$

Now, we can prove our main Theorem 9 for the section.

*Proof of Theorem 9.* From Lemma 14 we know that our algorithm is 5 consistent and so we have a bound on the connection cost (Lemma 1). To bound the facility cost we note that Lemma 15 implies that the bound on the facility cost is directly implied by a bound on the expected level of any vertex at the end of the algorithm. That is bounded in expectation by $O(\log \log \Delta)$ by Lemma 18. $\square$

**Dependence on the aspect ratio.** We emphasize that the above bound holds for arbitrary metric spaces. In particular, even if the aspect ratio $\Delta_t$ changes over time, the competitive ratio does not degrade. Furthermore, using the reduction in Section E, the dependence on $\Delta$ can be replaced by a dependence on $n$.

## C    PROOF OF THEOREM 11 AND THEOREM 12

**Theorem 11.** *There is an algorithm for the uniform learning-augmented facility location problem that at time $t$ produces an online solution with cost $O(\min\{\log(k + 1) \; \textbf{dynamic}_t, \frac{\log t}{\log \log t} \; \textbf{opt}_t\})$.*

*Proof.* The result is obtained by running the rounding algorithm presented in Section 3 on the fractional solution returned by the algorithm in Anand et al. (2022). Then, the result follows by combining Theorem 10 with Theorem 2. $\square$

**Theorem 12.** *There is an algorithm for the non-uniform learning-augmented facility location problem that at time $t$ produces an online solution with cost $O(\min\{\log \log \Delta \cdot \log(k + 1) \; \textbf{dynamic}_t, \frac{\log t}{\log \log t} \; \textbf{opt}_t\})$.*

*Proof.* We run the rounding algorithm presented in Section 4 on the fractional solution returned by the algorithm in Anand et al. (2022). Then, combining Theorem 10 with Theorem 9 gives a bound of $O(\log \log \Delta \cdot \min\{\log(k+1)\, \mathsf{dynamic}_t, \frac{\log t}{\log \log t}\, \mathsf{opt}_t\})$.

We now improve the robustness bound from $O(\log \log \Delta \cdot \frac{\log t}{\log \log t}\, \mathsf{opt}_t)$ to $O(\frac{\log t}{\log \log t}\, \mathsf{opt}_t)$. To do this, we use a standard technique of a combiner algorithm whose solution matches, up to a constant factor, the better of the rounded solution and that of an online algorithm without predictions. In our case, the latter algorithm is the $O(\frac{\log t}{\log \log t})$-competitive online facility location algorithm in Fotakis (2008).

We now give more details of how the two algorithms are combined to yield a robust algorithm. At any time $t$, compare the cost of the solutions (including both opening costs of facilities and connection costs of clients) of the two algorithms. For the algorithm that has the smaller cost, we open all facilities opened by that algorithm. So, clients can connect to their closest facilities given by this algorithm, and therefore, the total cost of opening these facilities and connecting clients is at most the total cost of the cheaper algorithm at time $t$.

For the other algorithm, we might have some facilities that are already open in the combiner algorithm from previous steps – we keep those facilities open but do not open any more facilities. Suppose $t' < t$ was the last time when the second algorithm was cheaper. Then, the open facilities in the combiner algorithm due to the second algorithm were already open in the second algorithm at time $t'$ (since the combiner algorithm does not open any new facilities from the more expensive algorithm). Now, note that the costs of the two algorithms are monotonically non-decreasing since the set of clients is monotonically increasing. This means the cost of the cheaper of the two algorithms is also monotonically non-decreasing. Thus, the cost of the cheaper algorithm at time $t$ is at least as much as the cost of the other algorithm at time $t'$ when it was the cheaper of the two algorithms. Hence, the total opening cost of the facilities in the second algorithm at time $t'$ is at most the current cost of the first algorithm at time $t$. It follows that the total cost of the combiner algorithm at any time $t$ is at most 2 times the cost of the cheaper of the two algorithms at time $t$. Since the algorithm without predictions has a competitive ratio of $O(\frac{\log t}{\log \log t})$, we get a robustness bound of $O(\frac{\log t}{\log \log t})$ using the combiner algorithm. $\square$

## D  LOWER BOUND ON APPROXIMATION GUARANTEE OF RANDOMIZED ALGORITHM

We show that the analysis of the randomized algorithm is asymptotically tight.

**Theorem 19.** *There is an instance for which the expected cost of the solution returned by the randomized algorithm is $\Omega(\log \log \Delta)$ times the cost of the fractional solution.*

The lower bound will consist of $10^d$ batches. The high-level intuition is that each batch will increase the level of (i.e., try to round) a special facility $u$ with probability at least $1/5^d$. This will allow us to prove that the randomized algorithm opens facility $u$ with a probability that is at least $d$ times the fractional opening value of $u$ given by the linear program. Then setting the opening cost of this facility so that it completely dominates the objective function will allow us to prove our lower bound of $\Omega(\log \log \Delta)$ because $d = \Omega(\log \log \Delta)$ in our construction. We remark that, as the randomized algorithm is oblivious to the points of the instance, we only describe in the construction how the fractional openings changes in the fractional solution that is rounded and omit the clients (that are irrelevant for the behavior of the randomized algorithm).

The metric is defined by the real line $\mathbb{R}$, and the distinguished vertex $u$ is positioned at the origin. The fractional value of $u$ is set to $\varepsilon > 0$, i.e., $y_u = \varepsilon$, deliberately chosen to be a tiny value. The cost of the special facility is $c(u) = M/\varepsilon$ and the cost of all other facilities is 0. We select $M \gg 10^{10^{2d}}$ sufficiently large so that the cost of the linear program solution is always dominated by $c(u) \cdot y_u = M$, i.e., the total connection cost is $o(M)$ (recall that the remaining opening cost is 0).

We now proceed with the description of the instance that yields the lower bound (see also Figure 2). For $b = 1, \ldots, 10^d$, the $b$th batch of arrivals is defined by the scale parameter $\gamma(b) = 10^{10^d(10^d - b)}$ and $d + 2$ facility locations that arrive during $d + 2$ time steps:

- First, center $c$ arrives at position $\gamma(b)$ with $y_c = 1/2 - 2\varepsilon$.
- During the next $d$ time steps, centers $v_1, v_2, \ldots, v_d$ arrive. Center $v_i$ is at position $\gamma(b)10^{d+1-i}$ and has $y_{v_i} = 1/5$.
- Finally, an extra center $c'$ colocated with $c$ arrives with $y_{c'} = \varepsilon$.

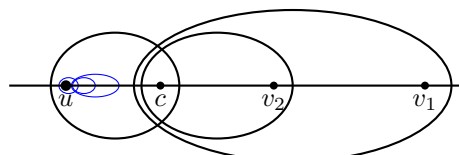

Figure 2: An illustration of the construction with $d = 2$. The ellipsoids depicted in black correspond to one batch and the blue correspond to the following batch. Due to the scaling factor $\gamma(\cdot)$ the different batches do not interact and by the geometrically decreasing radii of balls inside a batch we have that the critical balls of a batch are in order $(c, v_1), (c, v_2), \ldots, (c, v_d), (c, c', u)$ (the drawing is in logarithmic scale). The center $c'$ arriving last of a batch is colocated with $c$ and is not depicted.

By definition, the maximum distance in our construction is upper bounded by $\gamma(1) \cdot 10^d = \gamma(b) = 10^{10^d(10^d-b)} \cdot 10^d \leq 10^{10^{2d}}$ and the smallest distance is at least 1. Hence, by construction

$$\Delta \leq 10^{10^{2d}} \quad \text{and} \quad \log\log\Delta = O(d).$$

We now discuss the critical balls that are formed in our construction, which also gives valuable intuition for the construction. First, for a single batch, the critical balls that are formed (only using centers from that batch and $u$) are in order

$$(c, v_1), (c, v_2), \ldots, (c, v_d) \text{ and finally } (c, c', u).$$

(Here, we identify the balls with the locations that they contain for notational simplicty.) This is because the placement of the centers $c$ and $c'$ to be at position $\gamma(b)$ for batch $b$ and $v_i$ to be at position $\gamma(b) \cdot 10^{d+1-i}$ ensures that the radii of the above balls are rapidly decreasing. Specifically, when $v_i$ arrives, any ball $B$ containing centers from this batch with $y(B) \geq 1/2$ must either contain $c$ and $v_i$ or $v_i$ and $v_j$ with $j < i$. By their placement, the radii of the ball $(c, v_i)$ is significantly smaller than any ball containing $v_i$ and $v_j$. Additionally, it is significantly smaller than the previous critical balls which makes it critical. Finally, when $c'$ arrives, we have the critical ball of radii $\gamma(b)$ that contains $(c, c', u)$ (which now thanks to the arrival of $c'$ has $y$-value $1/2$). Repeating the same arguments (using that $\gamma(b+1) \ll \gamma(b)$), we can also conclude that there is no critical ball that contains two centers from two different batches.

We now proceed to analyze the random decisions of the algorithm. For simplicity and without loss of generality, we perform the analysis for $d \geq 10$. We say that a batch is *successful*, if the algorithm opens facilities $v_1, \ldots, v_d$ when considering critical balls $(c, v_1), \ldots, (c, v_d)$, i.e., it does *not* open center $c$.

**Lemma 20.** *There are at least $d$ successful batches with probability at least $1/2$.*

*Proof.* Let $p$ be the probability that a batch is successful. We first prove that $p \geq (1/5)^d$. When $v_1$ arrives, no center has been opened in the ball containing $(c, v_1)$ due to the batch scale factor $\gamma(\cdot)$. The algorithm opens $v_1$ with probability $y_{v_1}/(y_{v_1} + y_c) \geq y_{v_1} = 1/5$. Now, assuming the algorithm opens $v_1$, the same arguments say that the algorithm opens $v_2$ with probability at least $1/5$. That $p \geq (1/5)^d$ then follows by repeating the argument for $v_3, \ldots, v_d$.

The total number of batches are $10^d$ and so the expected number of successful batches are $10^d/5^d = 2^d$. The statement now follows via a standard Chernoff bound using that the success of different batches are independent events. For simplicity and completeness, we include a direct argument. The probability that exactly $\ell$ batches are successful is

$$\binom{10^d}{\ell} p^\ell (1-p)^{10^d - \ell} \leq \binom{10^d}{\ell} (1 - (1/5)^d)^{10^d}.$$

We have that $\sum_{\ell=0}^{d} \binom{10^d}{\ell} \leq (d+1)\binom{10^d}{\ell} \leq (d+1)10^{d^2}$. At the same time $(1-(1/5)^d)^{10^d} \leq 1/e^{10^d/5^d} = 1/e^{2^d}$. The statement now follows since $(d+1) \cdot 10^{d^2}/e^{2^d} \leq 1/2$ where we use that $d \geq 10$. $\qquad\square$

Now consider the case when there are at least $d$ batches that are successful. Consider the $d$ first such successful batches and let $E_i$ be the event that the randomized algorithm opens center $u$ when considering the $i$th of these batches.

**Lemma 21.** *We have* $\Pr[E_1 \vee E_2 \vee \cdots \vee E_d] = 2d \cdot \varepsilon$.

*Proof.* We have

$$\Pr[E_1 \vee E_2 \vee \cdots \vee E_d] = \sum_{i=1}^{d} \Pr[E_i \mid \neg E_1, \ldots, \neg E_{i-1}]$$

The lemma follows by arguing $\Pr[E_i \mid \neg E_1, \ldots, \neg E_{i-1}] = 2\varepsilon$. As the batch corresponding to event $E_i$ is successful, the facility $c$ is at level $d$ after the critical balls $(c, v_1), \ldots, (c, v_d)$. Therefore as $u$ has been considered $i-1 < d$ times at this point, i.e, is at level less than $d$, we have that the level of the ball $(u, c, c')$ is at most $d$. Therefore, the probability that $u$ is opened by the randomized algorithm equals $y_u/(y_u + y_c + y_{c'}) = 2y_u = 2\varepsilon$. Having proved that $\Pr[E_i \mid \neg E_1, \ldots, \neg E_{i-1}] = 2\varepsilon$, the lemma follows by the sum. $\qquad\square$

We have that the there are $d$ successful batches with probability at least $1/2$. If that holds, the probability that the randomized algorithm opens $u$, which incurs a cost of $M/\varepsilon$, is at least $2d\varepsilon$. It follows that the expected cost of the randomized algorithm is at least $d \cdot M$, which as noted above is $\Omega(\log \log \Delta)$ times the cost of the fractional LP solution. This completes the proof of Theorem 19.

# E  REDUCTION FROM $O(\log \log \Delta)$ TO $O(\log \log n)$

Recall that we have shown that the expected cost of the randomized algorithm is $O(\log \log \Delta)$ times that of the fractional solution. We now show that a slight modification of the algorithm changes the expected cost to $O(\log \log n)$ times the fractional cost, which might be more desirable if the number of vertices is small compared to the aspect ratio of the metric space.

By scaling, let us assume that the minimum distance between two non-identical vertices in the metric space is at least 1 and at most $\Delta$. (Note that for notational convenience, we created copies of identical vertices if multiple clients appear at the same vertex over time. However, this was simply a notational change, i.e., the competitive ratio of the algorithm is not affected if we switch back to a notation where distinct vertices are not co-located but the number of clients at a vertex can increase over time. We will take this latter view in this reduction.)

We first describe the reduction in the offline case, and then adapt it to the online setting. Our basic idea is to merge pairs of vertices that are within a distance of $\mathsf{opt}/n^2$, where $\mathsf{opt}$ is the cost of the fractional solution solution and $n$ is the total number of clients. To merge of a pair of vertices, consider a complete graph on all the vertices where the length of every edge is their pairwise distance. Now, change the length of the edge connecting the two vertices being merged to $0$ and recompute all pairwise distances in the metric space as the shortest path distances in the modified graph. Then, unify the two merged vertices into a single vertex whose opening cost is the smaller among the two merged vertices. We repeatedly perform this step of merging vertex pairs at a mutual distance of $\mathsf{opt}/n^2$ or less unless no such pair is left. (The precise order of the mergers is unimportant.) Note that as a result of these mergers, the following happen:

- The minimum distance between any pair of vertices in the modified metric space is $\geq \mathsf{opt}/n^2$.
- The difference in the distances between any pair of vertices in the original and modified metric spaces (in the modified metric space, each vertex represents multiple merged vertices of the original metric space) is at most $\mathsf{opt}/n$. This is because any path in the original metric space constitutes at most $|V| - 1$ merged pairs, where $|V| \leq n$ since each vertex has at least one client (we assume wlog that every vertex has at least one client). Therefore, the additional connection cost paid by a client in the original metric space compared to the modified one is $\leq \mathsf{opt}/n$, which adds up to $\leq \mathsf{opt}$ over all the $n$ clients.

Next, we decrease the maximum distance between any vertex pair. We define an unweighted graph on the vertices where each vertex pair that is at a distance of $\leq 2\mathsf{opt}$ is connected by an edge. Then, we identify the connected components of this auxiliary graph, and create a separate metric space for each connected component. This results in the following:

- The maximum distance between any pair of vertices in any of the individual metric spaces is $\leq 2(|V| - 1) \cdot \mathsf{opt} \leq 2n \cdot \mathsf{opt}$.

- For any client, the fractional solution connects at least half of the client within its component since connections across different components cost $> 2\mathsf{opt}$. Hence, it suffices for the algorithm to operate independently on each metric space by incurring a constant factor overhead in cost compared to the fractional solution.

Overall, these two steps ensure that the aspect ratio of the metric space on which the algorithm operates is at most $\mathrm{poly}(n)$, which in turn establishes an expected cost of $O(\log \log n)$ times the fractional cost for the randomized algorithm.

The above discussion only holds for an offline transformation to reduce $O(\log \log \Delta)$ to $O(\log \log n)$. In the online case, the first complication is that the value of $\mathsf{opt}$ is not known in advance and increases over time. We define a series of epochs, where an epoch ends when the value of $\mathsf{opt}$ doubles with respect to that at the beginning of the epoch. Suppose $\mathsf{opt}_t$ is the value of $\mathsf{opt}$ at time $t$ when an epoch starts. Then, we perform the transformations given above with $\mathsf{opt} = 2\mathsf{opt}_t$ so that $\mathsf{opt}$ remains an upper bound on the actual fractional cost throughout the epoch. Moreover, $\mathsf{opt}$ doubles in consecutive epochs thereby ensuring that the sum of $\mathsf{opt}$ across all the epochs is at most the final fractional cost times a constant factor.

In the rest of the discussion, we describe the reduction within a single epoch. The complication is that the number of clients $n$ also changes over time, which affects the parameters of the transformation. To handle this, we partition an epoch into a series of phases. A phase that starts when the number of clients is $n_t$ ends when the number of clients increases to $n_t^2$. (If the containing epoch ends before the end of a phase, then we start a new epoch and a new phase within that epoch.) Throughout the phase, we use $n = n_t^2$ in the above transformation, which ensures that $n$ is an upper bound on the actual number of clients. The expected cost in a phase is the fractional cost times $O(\log \log n) = O(\log \log n_t)$. Summing over all the phases in an epoch, we get $O(\log \log n)$ times the fractional cost at the end of the epoch, where $n$ is the number of clients in the epoch.

This completes the description of the overall reduction.

## F  RUNNING TIME ANALYSIS

Let $n$ be the number of vertices (clients and potential facilities) revealed up to time $t$. In a general metric space, we only have access to a pairwise distance matrix. The meaningful radii for any ball centered at a vertex are exactly the distances to the other $n - 1$ vertices. Therefore, at any time $t$, there are $O(n^2)$ distinct balls to track.

### F.1  DETERMINISTIC ROUNDING ALGORITHM (UNIFORM COSTS)

The computational bottleneck for the algorithm is finding the minimum radius balls that satisfy the required conditions.

Evaluating Condition A: The algorithm must find a ball $B(v, R)$ of minimum radius where the total fractional mass is at least $1/2$, and the closest open facility is at a distance $\geq 4R$.

Evaluating Conditions B and C: It then recursively searches for overlapping balls $B(u, r)$ and $B(w, \rho)$ with minimum radii satisfying specific mass and distance thresholds.

Naively, checking the fractional mass and the distance to the nearest open facility for all $O(n^2)$ possible balls takes $O(n)$ time per ball, leading to an $O(n^3)$ update time per step. However, if by maintaining a sorted arrays of distances from each vertex and by using a prefix sums for the fractional masses, we can update and evaluate the valid balls for a single center in $O(n)$ time.

This reduces a global scan to $O(n^2)$ time (after preprocessing), and yields a polynomial-time implementation overall.

### F.2 RANDOMIZED ROUNDING ALGORITHM (NON-UNIFORM COSTS)

The algorithm is computationally heavier because it requires global sorting and sequential overlap checking.

Lexicographic Sorting: The algorithm enforces a strict lexicographic order on all balls $B(v, R)$ based on arrival time, radius, and vertex id. Maintaining the sorted order of $O(n^2)$ balls takes $O(n^2 \log n)$ time.

Identifying Critical Balls: A ball is deemed "critical" if its mass is $\geq 1/2$ and it does not overlap with any previously ordered critical ball of radius $\leq 2R$. To determine this, the algorithm must iterate through the $O(n^2)$ sorted balls. Overlap checking can be implemented in $O(n)$ time per ball by maintaining, for each vertex $u$, the minimum radius of any previously-declared critical ball containing $u$; then $B(v, R)$ overlaps such a ball iff it contains some $u$ with stored value $\leq 2R$. This yields an $O(n^3)$ worst-case bound per online step for identifying critical balls.

Level Calculation and Rounding: Calculating the dynamic level $\ell(B)$ for a critical ball requires summing masses based on individual vertex levels. Executing the dependent and independent probability flips for vertices within the ball takes $O(n)$ time per critical ball.

Overall, the overlap-checking bottleneck brings the worst-case runtime to $O(n^3)$ per online time step.