# OpenReview forum: "Online Rounding and Learning Augmented Algorithms for Facility Location"
_ICLR.cc/2026/Conference — ICLR 2026 Poster_

### Official Review · Reviewer_jk99 · 2025-10-29

**Soundness:** 3
**Presentation:** 2
**Contribution:** 2
**Rating:** 6
**Confidence:** 3

**Summary:**

The paper presents the first online rounding algorithms that convert fractional online facility-location solutions into integral ones while preserving competitive theoretical guarantees. The main results include: (1) a deterministic constant-factor rounding algorithm for uniform opening costs, and (2) a randomized rounding algorithm for non-uniform costs with $O(\log\log\Delta)$ expected loss, where $\Delta$ is the aspect ratio of the given instance. These rounding schemes are then plugged into prior fractional, learning-augmented (multiple-predictions) algorithms to yield the first integral solutions with $O(\log(k+1))$ (uniform) and $O(\log\log\Delta\log(k+1))$ (non-uniform) consistency, together with $O(\log t/\log\log t)$ robustness via a combiner. A lower bound shows the $O(\log\log \Delta)$ dependence is asymptotically tight for the randomized scheme, supporting the claim that the main hardness is obtaining a strong fractional solution rather than rounding it.

**Strengths:**

The uniform-cost algorithm is 4-consistent with a constant-factor competitive cost. For non-uniform costs, the algorithm is 5-consistent and can control the expected facility cost via a vertex-level scheme, achieving $O(\log\log \Delta )$. The resulting level/critical-ball construction is clear and metric-aware. Combined with the fractional learning-augmented algorithm of Anand et al. (2022), the rounding process yields the first integral multiple-prediction results with nearly-tight consistency and robustness guarantees. A matching $\Omega(\log\log \Delta)$ lower bound shows this dependence is unavoidable for randomized rounding, supporting the claim that the main hardness lies in finding a strong fractional solution, not in rounding. Finally, the appendix gives a reduction that trades $\log\log \Delta$ for $\log\log n$, which is useful when $\Delta$ is large with moderate data size $n$.

**Weaknesses:**

The main weakness can be summarized as follows.

1. For non-uniform costs, the guarantees scale with the metric aspect ratio $\Delta$ (or, via reduction, with $n$), which can be large in practice.

2. The robustness improvement chooses the cheaper of two online solutions by opening all its facilities, while facilities opened by the other solution remain. This preserves guarantees but can monotonically increase the number of open facilities over time; the paper does not analyze recourse or stability beyond competitive cost.

3. It is suggested that the paper should clarify the novelty, where the proposed  is the first online rounding algorithms for metric facility location, not the first results on online facility location overall.

4. Because the three-level ball scheme is central, giving an explicit example before Conditions A/B/C would make the presentation clearer.

**Questions:**

1. It is unclear for me that why the online setting is the right lens (irrevocable decisions under partial information), and why rounding—rather than runtime—is the key bottleneck.

2. Could you instantiate the model with a simple scenario (e.g., two competing predictors) and show how your rounding plus the combiner behaves under accurate vs. adversarial advice?

3.  Can the randomized rounding guarantee be made $O(\log\log n)$ in the main text (not only via Appendix E’s reduction), or do structural barriers preclude this?

---

> ### Author Response · Authors · 2025-11-19
>
> We thank the reviewer for the detailed comments and insightful suggestions.
>
> Response to Weaknesses:
>
> - We show in Appendix E that the bound only grows as $O(\log \log n)$. For the non-uniform case, the competitive ratio grows very slowly with $n$ as $O(\log \log n)$. To give a sense of the growth, the value of $\ln \ln n$ is around $3$ when $n$ is a billion. We will clarify this point in the final version and explicitly mention that our results do not depend on the aspect ratio.
>
> - This paper focuses on the online setting where the solution is required to be monotone(as in previous work), i.e., there is no recourse and open facilities cannot be closed. If recourse is permitted and facilities that were previously open can be closed, then the robust algorithm can simply close the previously open facilities when switching from one solution to the other one. Since the number of open facilities is at most the number of points in the metric space, the amortized recourse is at most $1$. We will include a discussion of the recourse setting in the paper.
>
> - As suggested by the reviewer, we will clarify that this is the first algorithm for online rounding for metric facility location and not the first algorithm for online metric facility location in the paper.
>
> - Thanks for the suggestion. We will give an explicit example with the three balls in the rounding algorithm for uniform costs to help illustrate our technique.
>
>
> Response to Questions:
>
> - Runtime and quality of the solution are orthogonal parameters for an algorithm. In online settings, the focus is typically on the quality of the solution captured by the competitive ratio. We note that there is a wealth of literature in ML-augmented online algorithms that have appeared in ICML, Neurips, and ICLR for a large variety of problems in recent years (see the papers cited in the related work section of our paper). All these papers focus on the competitive ratio of the algorithm as against runtime, which is also the case in this paper. This is to understand the information-theoretic challenges of the problem in the online setting that are separate from computational challenges. We note that our algorithm runs in polynomial time. We will include a short runtime analysis in the paper.
>
> - We will provide an instantiation for the scenario described by the reviewer. We note that since our algorithm uses a fractional solution as input, this simulation would necessarily need to involve prior work such as Anand et al. (ICML 2022) that produces a fractional solution for this scenario, which can then be rounded online by our algorithm.
>
> - We will clarify that the bound obtained in Appendix E holds for all metric spaces and is completely general. We will state this in the main body of the paper.

---

> ### Comment · Reviewer_jk99 · 2025-11-26
> **Response to the Authors**
>
> Thanks for the responses. Most of my concerns have been addressed, and I prefer to keep my original evaluations at this stage.

---

### Official Review · Reviewer_TNrV · 2025-10-31

**Soundness:** 2
**Presentation:** 1
**Contribution:** 2
**Rating:** 4
**Confidence:** 5

**Summary:**

This paper studies the online facility location problem under the learning-augmented setting, aiming to bridge the gap between fractional and integral solutions. The authors propose two online rounding algorithms-- a deterministic algorithm for the uniform-cost case achieving a constant approximation ratio and a randomized algorithm for the non-uniform-cost case achieving an $O(\log \log \Delta)$ approximation.

These algorithms can be combined with existing fractional solutions (Anand et al., 2022) to yield near-optimal integral algorithms for the learning-augmented facility location problem. The work establishes that the main difficulty of the classic online facility location lies in computing high-quality fractional solutions rather than rounding them.

**Strengths:**

1. The results provide a conceptual bridge between fractional and integral formulations, advancing the theoretical understanding of online algorithms.

2. The deterministic algorithm achieves constant-factor guarantees, while the randomized version achieves $O(\log \log \Delta)$, improving over previous bounds.

3. The paper tackles an open problem by presenting the first online rounding algorithms for the facility location problem in both uniform and non-uniform cases.

4. The approach extends learning-augmented facility location to integral settings with nearly tight consistency–robustness trade-offs.

**Weaknesses:**

1. Although the paper successfully extends the known fractional results to integral settings, the achieved approximation and consistency–robustness bounds do not improve upon the theoretical lower bounds established in prior works (Almanza et al., 2021; Anand et al., 2022). In other words, the results are asymptotically tight but not stronger than existing bounds, which somewhat limits the theoretical novelty of the contribution.

2. The technical exposition, while mathematically sound, is dense and could benefit from clearer intuition or illustrative examples of the rounding procedure.

3. The applicability of the rounding framework to other online covering problems (e.g., online set cover, k-median, or caching) is not discussed, which limits the broader impact of the work.

4. Although the theoretical results close an existing gap between fractional and integral formulations, the technical novelty of the algorithms themselves seems incremental. Many arguments (e.g., in Lemma 1 and Theorem 2) follow standard geometric consistency proofs, and the randomized rounding procedure closely resembles existing metric-dependent rounding techniques. As a result, the contribution may feel more like a careful synthesis of known tools than a fundamentally new technical breakthrough.

5. The paper is entirely theoretical. There is no experimental validation or numerical evidence illustrating the practical behavior of the algorithms or how large the hidden constants are in real settings.

**Questions:**

1. Could the authors provide empirical or synthetic experiments to demonstrate the actual performance or confirm the tightness of the constants?

2. Is it possible to adapt the rounding framework to directly construct improved fractional algorithms or to other related online optimization problems?

3. How sensitive is the performance to the quality of the fractional input—e.g., if the fractional approximation is only moderately good?

4. The paper argues that the rounding stage is no longer the main source of difficulty in online facility location. Could the authors provide further justification or empirical evidence supporting this claim—for instance, by showing that different fractional baselines yield similar integral performance after rounding?

---

> ### Author Response · Authors · 2025-11-19
>
> We thank the reviewer for the detailed comments and questions.
>
> Response to Weaknesses:
>
> The bounds achieved in this paper improve the prior bounds in Almanza et al. and Anand et al. in the following ways:
> - Almanza et al.’s result only applies to the case of uniform costs. They do not have a result for the more general non-uniform costs, for which our result applies.
> - Almanza et al.‘s result for the uniform case loses a factor of $O(\log \log n)$ in comparison to our result for the uniform cost case. So we asymptotically improve on it.
> - Anand et al. do not provide an integral solution for either uniform or non-uniform costs. They only give a fractional solution. In contrast, we give an integral solution for both cases. Note that in practice in most scenarios an integral solution for the problem is needed.
>
> We will add more examples and intuition to improve the readability of the technical results. Thanks for the suggestions.
>
> For online set cover, there is a lower bound of $O(\log n)$ on any online rounding algorithm since the best achievable fractional and integral competitive ratios differ by this factor. Therefore, the results in this paper that achieve a rounding loss of $O(1)$ and $O(\log \log n)$ for uniform and non-uniform costs respectively are not replicable for online set cover. Indeed, online set cover is equivalent to non-metric versions of the facility location problem, while our results are specifically leveraged on metric properties. Since clustering problems are typically defined on metric spaces, we believe this distinction is crucial in this context.
> For $k$-median, the natural way to define an online problem is by going to the Lagrangian relaxation, which is exactly the online facility location that we consider in this paper. If one were to strictly enforce the constraint that there are only $k$ clusters, then it is easy to show that no online algorithm can be competitive.
> For online caching, there are existing online rounding algorithms that only lose an $O(1)$ factor.
>
> We Would kindly ask the reviewer to clarify what they mean by “standard geometric consistency proofs” and “metric-dependent rounding techniques”. We are not aware of any prior work that provides a rounding algorithm that leverages metric or geometric properties in the online setting. In fact, it is worth noticing that the literature on online rounding is rather scarce at the moment. There are such works in the offline setting, but the techniques used in the offline setting are very different from the techniques we develop in the online setting in this paper. For instance, in offline facility location, the standard rounding technique is to use a two-phase procedure where the first phase tentatively opens facilities and the second phase reassigns clients to a subset of permanently opened facilities. Such a two-phase technique cannot be used in the online setting since a prior decision of opening  a facility cannot be revisited later in the algorithm.
>
> Our contribution is primarily theoretical in nature: we obtain the first non-trivial online rounding algorithm for facility location, which is an important problem in clustering, and give precise bounds on its competitive ratio. It would be interesting to evaluate the performance of our algorithm empirically, but similar to the existing literature that we build on (Anand at al. ICML 2022), we have not performed experiments in this paper.
>
> Response to Questions:
>
> - Please see 5th paragraph in response to weaknesses.
> - Please see 3rd paragraph in response to weaknesses.
> - The online rounding algorithm is not sensitive at all to the quality of the fractional solution. In other words, given any fractional solution, it produces an integral solution that is within $O(1)$ (for uniform costs) or $O(\log \log n)$ (for non-uniform costs) of the fractional cost. We also note that if the quality of the advice or the fractional input is low one always recovers a solution that matches asymptotically the quality of the best possible online algorithm solution for the facility location problem.
> - The rounding stage is no longer the bottleneck because the overall competitive ratio is now entirely based on the quality of the fractional solution. Instead of having to design new fractional algorithms and also rounding algorithms, it now suffices to simply design a fractional algorithm and use our rounding algorithm as a black box to produce a better integer algorithm. We will clarify this in the final version.

---

> ### Comment · Reviewer_TNrV · 2025-11-26
> **Response to the Authors**
>
> Thank you for the thoughtful rebuttal and clarifications. The response helps distinguish the new bounds from those in prior work, clarifies why the analysis does not follow standard geometric-consistency or metric-dependent rounding templates, and explains better how the competitive ratio depends (or does not depend) on the quality of the fractional solution. However, the contribution still feels largely incremental and primarily technical, with limited intuition about when the improved bounds are practically meaningful, and the lack of any empirical evaluation leaves the real impact of the proposed rounding algorithm somewhat unclear. For these reasons, I would like to maintain my original overall score.

---

### Official Review · Reviewer_CpWi · 2025-11-01

**Soundness:** 2
**Presentation:** 3
**Contribution:** 3
**Rating:** 6
**Confidence:** 3

**Summary:**

This paper discusses rounding algorithms for the fractional facility location problem under both the cost structures, uniform and non-uniform. That is, when the cost of establishing a new facility is either constant or varies across facility locations. It presents interesting theoretical results that address one of the key open problems of facility location topic. Authors propose online learning algorithms that leverage the solution of the fractional problem to construct solutions for the corresponding integral versions. The algorithms are well-motivated, clearly explained, and supported by sound theoretical reasoning.

**Strengths:**

The paper presents proposes a novel solution for facility location problem. The requests arrive sequentially and hence this is related to the  clustering (unsupervised learning) problem. The performance measure is approximation ratio. The key idea is to transform the fractional solutions of the online facility location problem into feasible integral solutions. The authors build upon and extend prior approaches that address several key limitations, achieving a better approximation ratio.


Authors consider two main settings: one where the facility setup cost is uniform across all locations, and another where the costs vary by vertex (the non-uniform cost structure). This broader treatment is interesting as the model is more realistic and relevant to practical scenarios.


The paper has strong theoretical depth; the theorems and their proofs developed nicely. While the discussion connects partially to existing literature, the algorithmic descriptions are particularly detailed and well explained. Important notions such as critical balls, randomized rounding, and the role of aspect ratios in determining approximation guarantees are well defined. These explanations enhance the readability of the technical sections; interested readers, who may not be into this area, may also be able to follow.

**Weaknesses:**

The paper is theoretical in tis nature, focusing on proving approximation bounds and analyzing the properties of rounding algorithms. While the theoretical contributions are strong, the practical relevance particularly in learning-augmented settings could be strengthened through empirical validation. Incorporating simulations or experiments on synthetic or real-world datasets would help demonstrate the applicability of the proposed methods beyond the purely theoretical framework.

Next, several assumptions made in the paper, such as the preservation of fractional mass and the existence of specific geometric or metric properties (e.g., aspect ratio bounds), are not well justified. It is unclear whether these are standard assumptions in prior literature or newly introduced for tractability. If these are common, the authors should explicitly cite supporting works; if they are novel, a more thorough discussion or justification of their necessity and implications is required. As of now, some of these assumptions appear tailored to make the analysis feasible rather than naturally arising from the problem structure. Comments by the Authors are desirable.

The paper also references the open challenge of achieving an approximation factor of O(log /log log K) in the non-uniform cost case. However, this result is asymptotic, and this limitation is not highlighted in the discussion. It is very desirable if Authors explicitly acknowledge this gap and outline possible directions for bridging this gap.

As far as the presentation is concerned, several definitions such as critical balls, levels, and probabilistic rounding steps are dense, making it hard to parse. Many readers would like to have easy to explain examples or see illustrations (schematic diagrams); they could greatly enhance the readability for the readers that are less familiar with the literature.

Finally, while the derived bounds involving factors like (log \Delta) are theoretically interesting, the tightness of these bounds is not well discussed. Clarifying whether these bounds are provably near-optimal or if there is potential for improvement would add depth to the technical discussion. Computational illustrations help here.

**Questions:**

In addition to the above, I would suggest some clarifications and further discussion on the following points:

In the absence of computational evaluations, It is not clear to me how the proposed algorithms perform on small-and large-scale instances, especially those with high aspect ratios or complex metric spaces. At least small scale (initial) experimental results evaluating the computational complexity and scalability of the proposed rounding algorithms would be useful.


Are there specific optimization techniques that Authors recommend for implementing the algorithms efficiently in practice?
The analysis assumes that the fractional mass y^t_v​ remains fixed during the rounding process or can be approximated as such via vertex splitting. Could you comment on real-world scenarios where this assumption may not hold strictly? How would potential violations of this assumption affect the validity or tightness of the approximation guarantees?


The derived bounds naturally depend on the aspect ratio of the metric space, \Delta_t​. In dynamic or evolving settings where this ratio may vary significantly over time, how sensitive are the theoretical guarantees to such changes?

In the absence of experimental results, it remains unclear how the proposed methods compare with existing online facility location algorithms, such as those by Meyerson or more recent learning-augmented approaches. A qualitative or theoretical comparison in terms of approximation factors and operational complexity would help position this work within the broader literature.

---

> ### Author Response · Authors · 2025-11-19
>
> We thank the reviewer for the detailed feedback and constructive comments.
>
> Response to Weaknesses
>
> We agree that our contribution is primarily theoretical: we obtain the first non-trivial online rounding algorithm for facility location, which is an important problem in clustering, and give precise bounds on its competitive ratio. We also think that it would be interesting to evaluate the performance of our algorithm empirically, but similarly to the existing literature that we build on (for example Anand at al. ICML 2022), we have not performed experiments in this paper.
>
> We would like to clarify that preservation of fractional mass at any point in the metric space is not an assumption; it holds without loss of generality. This is because having two points in the metric space with a distance of 0 between them is equivalent to having a single point with the sum of their fractional values. Similarly, there is no assumption in the paper about the aspect ratio of the metric space. As shown in Appendix E, the dependence on aspect ratio $\Delta$ can be replaced by a dependence on $n$. We would like to emphasize that our results hold for all metric spaces and for all instances of facility location. We will clarify this point in the final version and make more explicit references to Appendix E.
>
> Unfortunately we are not sure that we understand the third suggested weaknesses, it would be great if the reviewer could clarify what they mean by “open challenge of O(log/log log k)” for the non-uniform case. Note that prior to our work, there was no non-trivial online rounding algorithm known for either the uniform and non-uniform case. Our work presents online rounding algorithms with competitive ratio $O(1)$ and $O(\log \log k)$ for the uniform and non-uniform cases respectively. Therefore, there is no logarithmic gap for the non-uniform case given our work. It is plausible that the bound for the non-uniform case can be improved from $O(\log \log k)$ to $O(1)$. We showed in the appendix that the analysis of our current algorithm is tight, which implies that a new algorithm is required for the better bound. The main difference between the uniform and non-uniform cases is that the cost of opening a center at one location of the metric space can be charged to the fractional cost in a different part of the metric space in the uniform case, but this cannot be done in the non-uniform case. We will include a discussion on this in the paper.
>
> Thanks for the suggestions. We will include an illustration giving an example of the key concepts in our algorithm and analyses such as critical balls and the sequence of steps in the randomized rounding algorithm.
>
> As shown in Appendix E, the dependence on the aspect ratio $\Delta$ in our bounds can be replaced by a dependence on $n$. In other words, we show that for metric spaces that have unbounded $\Delta$, the competitive ratio does not need to depend on $\Delta$.

---

> > ### Author Response · Authors · 2025-11-19
> >
> > Response to Questions:
> >
> > - In the absence of computational evaluations, It is not clear to me how the proposed algorithms perform on small-and large-scale instances, especially those with high aspect ratios or complex metric spaces
> >
> > We emphasize that the dependence on the aspect ratio is not required, and that our bounds hold for any instance of facility location on any metric space, irrespective of the complexity of the metric space. As mentioned above, our contribution is primarily theoretical. While it would be interesting to evaluate the performance of our algorithm empirically, we have not performed experiments in this paper, which is also the case for the existing literature that we build on (Anand at al. ICML 2022).
> >
> > - Are there specific optimization techniques that Authors recommend for implementing the algorithms efficiently in practice? [...]
> >
> > We would like to clarify that the fact that the fractional value at a point remains constant over time by duplicating points at the same location is not an assumption. It can be implemented without loss of generality in any metric space since having two points that are distance $0$ from one another is exactly equivalent to having a single point with a fractional mass equal to the sum of the two points. This equivalence holds irrespective of the metric space, and hence, our result holds for every metric space. Furthermore, it does not impact the efficiency of the algorithm. We will clarify this point in the final version of the paper.
> >
> > - The derived bounds naturally depend on the aspect ratio of the metric space, \Delta_t​. In dynamic or evolving settings where this ratio may vary significantly over time, how sensitive are the theoretical guarantees to such changes?
> >
> > Thanks for raising this important point. As shown in Appendix E, the dependence on the aspect ratio $\Delta$ is not required at all, and can be replaced by a dependence on $n$. Therefore, even if $\Delta$ changes over time, the competitive ratio of our algorithm does not degrade over time. We will clarify this point in the final version of the paper.
> >
> > - [...] A qualitative or theoretical comparison in terms of approximation factors and operational complexity would help position this work within the broader literature.
> >
> > Thanks for the suggestion. We will emphasize and clarify the improvements obtained by our work to learning-augmented online facility location, both for the settings considered by Almanza et al.(Neurips 2021) and Anand et al. (ICML 2022), in the paper. In particular, the Almanza et al. paper does not apply to the non-uniform case and the Anand et al. paper only produces a fractional solution. Therefore, we obtain the first (integral) online facility location algorithm with multiple predictions with non-uniform costs. Even for uniform costs, the Almanza et al. paper loses a factor of $O(\log \log n)$ in the integral solution compared to the fractional solution, which we improve to $O(1)$ in this paper. We will add a table to the paper to make this comparison more evident in the final version.

---

### Official Review · Reviewer_9sc3 · 2025-11-05

**Soundness:** 3
**Presentation:** 3
**Contribution:** 3
**Rating:** 8
**Confidence:** 4

**Summary:**

This paper studies the Facility Location problem, a fundamental clustering task, in the context of online learning augmented algorithms. The authors identify a key gap in existing research: while strong, nearly tight bounds exist for the fractional version of the problem (where facilities can be partially opened), the integral version (where facilities are either fully open or closed) remains less understood. The paper's main contribution is to bridge this gap by introducing the first online rounding algorithms designed to convert a fractional solution into an integral one efficiently. Two algorithms are presented: first, a deterministic algorithm for the uniform cost case, in which all facilities have a cost of $1$, that achieves a constant $(O(1))$ cost overhead compared to the fractional solution, and second, a randomized algorithm for the non-uniform cost case that incurs an $O(\log\log\Delta)$ loss. By combining these rounding techniques with the known fractional algorithms of Anand et al. (2022), the paper derives the first integral learning-augmented facility-location algorithms with consistency and robustness guarantees that match the previously known ones for the fractional case.

**Strengths:**

One of the paper’s strengths is its novelty and solid theoretical contribution, as it introduces the first online rounding algorithms specifically for the facility location problem, essentially reducing the integral version of the problem to the fractional one. The paper is also well structured, with proofs that are intuitive and technically involved.

**Weaknesses:**

The paper title might be a bit misleading as the results are entirely on online rounding algorithms with applications to learning augmented algorithms. The results can be applied to existing learning augmented algorithms, but the novelty is only on the online rounding part.

**Questions:**

- Could you apply similar techniques to directly get an online learning augmented algorithm by treating the input fractional solution as the ML advice?
- You mention as future work finding a constant-factor rounding for the non-uniform case. What seems to be the main obstacle for that?

---

> ### Author Response · Authors · 2025-11-19
>
> Thank you for your supportive and constructive comments. We are glad that you liked the paper.
>
> Response to Weaknesses:
>
> Thank you for your suggestion about tweaking the title of the paper to highlight its key contribution of online rounding for facility location problems. We would be glad to adopt your suggestion and change(if possible) the title to “Online Rounding and Applications to Learning-Augmented Algorithms for Facility Location”.
>
> Response to Questions:
>
> - Could you apply similar techniques to directly get an online learning augmented algorithm by treating the input fractional solution as the ML advice?
>
> Thanks for the suggestion, we can indeed apply our results directly to a fractional advice and get similar results. This is true because our rounding algorithms can act directly on the advice and robustness can be obtained thanks to known results of Mahdian et al. [2012] and Meyerson [2001] as in Almanza [2021].
>
> - You mention as future work finding a constant-factor rounding for the non-uniform case. What seems to be the main obstacle for that?
>
> Thanks for your question. We showed in the appendix that the analysis of our current algorithm for the non-uniform case is tight, which implies that a new algorithm is required for the better bound of $O(1)$. The main difference between the uniform and non-uniform cases is that the cost of opening a center at one location of the metric space can be charged to the fractional cost in a different part of the metric space in the uniform case, but this cannot be done in the non-uniform case. We will include a discussion on this in the paper.
>
> References:
> - Adam Meyerson. Online facility location. In Proceedings of the 42nd IEEE Symposium on Foundations of Computer Science, FOCS ’01, page 426, USA, 2001. IEEE Computer Society. ISBN 0769513905.
> - Mohammad Mahdian, Hamid Nazerzadeh, and Amin Saberi. Online optimization with uncertain information. ACM Transactions on Algorithms (TALG), 8(1):1–29, 2012.
> - Matteo Almanza, Flavio Chierichetti, Silvio Lattanzi, Alessandro Panconesi, and Giuseppe Re. Online facility location with multiple advice. Advances in neural information processing systems, 34:4661–4673, 2021.

---

### Meta-Review · Area_Chair_ebn5 · 2026-01-07

**Summary:**

The paper provides a theoretical study of the facility location problem. The main new contribution is online algorithms for the integral version of the facility location problem, with applications to online facility location with machine-learned advice. The reviewers are generally positive about this work. Most of the concerns have been addressed in the rebuttal.

**Reviewer Concerns:**

Most concerns have been addressed. There is a remaining concern about the lack of empirical evaluation by Reviewer TNrV. However, since this work belongs to the field of algorithm theory, empirical evaluation is unnecessary.

**Reviewer Scores:**

Based on the existing discussions, it seems all the reviewers would have kept their original scores.

---

### Decision · Program_Chairs · 2026-01-26

Accept (Poster)